# Improved Adaptive Multipoint Optimal Minimum Entropy Deconvolution and Application on Bearing Fault Detection in Random Impulsive Noise Environments

**DOI:** 10.3390/e25081171

**Published:** 2023-08-06

**Authors:** Yu Wei, Yuanbo Xu, Yinlong Hou, Long Li

**Affiliations:** School of Automation, Xi’an University of Posts and Telecommunications, Xi’an 710121, China; xuyuanbo@xupt.edu.cn (Y.X.); houyinlong@xupt.edu.cn (Y.H.); lilong@xupt.edu.cn (L.L.)

**Keywords:** rolling element bearing, fault characteristic detection, multipoint optimal minimum entropy deconvolution, particle swarm optimization, envelope autocorrelation function, random impulsive noise

## Abstract

Random impulsive noise is a special kind of noise, which has strong impact features and random disturbances with large amplitude, short duration, and long intervals. This type of noise often displays nonGaussianity, while common background noise obeys Gaussian distribution. Hence, random impulsive noise greatly differs from common background noise, which renders many commonly used approaches in bearing fault diagnosis inapplicable. In this work, we explore the challenge of bearing fault detection in the presence of random impulsive noise. To deal with this issue, an improved adaptive multipoint optimal minimum entropy deconvolution (IAMOMED) is introduced. In this IAMOMED, an envelope autocorrelation function is used to automatically estimate the cyclic impulse period instead of setting an approximate period range. Moreover, the target vector in the original MOMED is rearranged to enhance its practical applicability. Finally, particle swarm optimization is employed to determine the optimal filter length for selection purposes. According to these improvements, IAMOMED is more suitable for detecting bearing fault features in the case of random impulsive noise when compared to the original MOMED. The contrast experiments demonstrate that the proposed IAMOMED technique is capable of effectively identifying fault characteristics from the vibration signal with strong random impulsive noise and, in addition, it can accurately diagnose the fault types. Thus, the proposed method provides an alternative fault detection tool for rotating machinery in the presence of random impulsive noise.

## 1. Introduction

Bearings are an omnipresent element in rotating machinery, playing an indispensable role in industrial production, processing, and manufacturing. Bearings are mainly used to support the main shaft and transmit torque within rotating machines [1]. In the event of local damage to bearings, noise, vibrations, and other abnormal phenomena will occur, and even lead to machine breakdown and accidents in serious cases [2]. Hence, delving into the realm of bearing condition monitoring and fault diagnosis holds immense significance.

In real–world applications, vibration signals extracted from damaged bearings typically exhibit a considerable amount of background noise due to complex vibration transmission paths, serious noise perturbations in the working environment, and multi–vibration source excitation [3]. In addition, the fault–related periodic impulses exhibit an exponentially decaying trend [4]. The weak periodic impulses are easily submerged by background noise. Revealing fault–related cyclic impulsive components from measured vibration signals is therefore of paramount importance for bearing fault diagnosis. The filtration of raw signals is one of the most commonly used methods. Fortunately, advanced techniques for filtering out background noise have reached a high level of sophistication in the field of bearing fault diagnosis, including wavelet transform [5,6,7], mode decomposition [8,9,10], deconvolution methods [11,12,13], and other filtering tools [14,15,16].

However, a special kind of noise, namely random impulsive noise, can also appear in certain engineering problems such as telephone wires, image processing, radar systems, and underwater acoustics. Random impulsive noise is clearly different from traditional background noise. Random impulsive noise often exhibits non–Gaussianity, while traditional background noise obeys Gaussian distribution. Furthermore, random impulsive noise shows strong impact features and has sudden disturbances with varying amplitude, short duration, and long intervals [17]. According to the above discussion, it is found that random impulsive noise is somewhat similar to fault–induced impulsive components. Both comprise a series of impulsive components, but random impulsive noise has a random distribution with varying amplitudes whereas fault–related impulsive components have a periodic distribution with a uniform amplitude. Unlike other engineering applications, distinguishing fault–induced cyclic impulses from random impulses poses a significant challenge in bearing fault diagnosis. Compared to Gaussian background noise, random impulsive noise can result in more severe signal corruption. Accordingly, the above–mentioned signal processing tools used for analyzing vibration signals may perform poorly or become ineffective [18,19].

The most popular method used for dealing with random impulsive noise in the literature is the informative frequency band (IFB) selector. The IFB selection methods are initially inspired by the fast kurtogram (FK) method proposed by Antoni [20]. FK is developed based on spectral kurtosis. Since kurtosis can measure the impulsiveness induced by local faults, the impulsive information can be selected at a narrow frequency band obtained by FK and filter out other interfering components. Nevertheless, kurtosis is highly susceptible to outliers, which may lead to detecting the impulsiveness of the random impulsive noise as opposed to the fault–related impulsive components. FK fails to accurately estimate the resonance frequency band in such situations. To better cope with random impulsive noise, the variations and extensions of kurtosis and alternative statistical indicators for impulsiveness detection have been widely employed as substitutes for kurtosis. Since the alpha-stable distribution is an effective measure for quantifying the overall impulsiveness of a fault signal, Yu et al. [21] proposed an alpha-stable-based IFB selector. Miao et al. [22] used the Gini index to replace kurtosis in designing the Gini-guided-gram method for bearing fault detection. A further use of the Gini index for extracting bearing faults is demonstrated in the research conducted by Wang et al. [23]. Antoni [24] proposed an IFB selector called Infogram, which substitutes kurtosis with negentropy. Hebda-Sobkowicz et al. [25] proposed a new IFB selection method based on the conditional variance statistic. In addition to these alternative statistical indicators, some updated versions of kurtosis are also utilized to develop the IFB selectors. Liao et al. [26] proposed a frequency domain multipoint kurtosis-based IFB selector named Fast Mkurgram for bearing fault diagnosis. Moshrefzadeh et al. [27] used the autocovariance function to propose an IFB selector named Autogram, which used the autocovariance function to obtain the kurtosis of the unbiased autocorrelation of the squared envelope of the signal, instead of that of the raw signal. Cheng et al. [28] proposed a feature-adaptive IFB selection method named improved envelope spectrum via candidate fault frequencies’ optimization-gram (IESCFFOgram). This IFB selection method serves as a suitable tool for detecting bearing faults in the absence of accurate fault characteristic frequencies. It is clearly found that the IFB selection techniques heavily rely on the performance of the statistical indicators. When working conditions and environments change dramatically, it is necessary to consider the relevance of statistical indicators. In fact, when the large non–cyclic impulsive noise appears in vibration signals then some of the IFB selectors based on statistical indicators may fail to estimate an accurate resonance frequency band (bandwidth and center frequency), resulting in the failure of the subsequently designed band-pass filter [17,25].

The IFB–based methods can be considered a type of filtering technique because the estimated parameters of the bandwidth and center frequency are employed to design a band–pass filter. As discussed previously, an inaccurate estimation of bandwidth and center frequency can render the designed band–pass filter ineffective. Therefore, we use an alternative idea to handle random impulsive noise in this paper. Periodic impulses are recognized as the most significant representation of bearing faults. The repetitive impulsive components look like an arithmetic sequence and the interval between two consecutive impulses remains constant. Obviously, identifying a series of regular impulses is more simple when two different groups of impulsive components coexist simultaneously. The idea is straightforward: locate each fault–induced impulse and then extract them from raw signals. Fortunately, the multipoint optimal minimum entropy deconvolution (MOMED) method is exactly in line with this concept. The MOMED method aims to find a target vector to define the location and weightings of the goal impulses to be deconvolved [29]. Thus, the MOMED method is similar to an inverse filter that can highlight the cyclic impulses rather than directly filtering out noise. However, the MOMED is highly dependent on the pre–defined range of the cyclic impulse period to highlight the fault–induced impulsive components, so it cannot automatically determine the fault–related period. Additionally, the performance of the MOMED method depends on the appropriate selection of the filter size. Increasing the filter length can improve the ability to enhance raw signals, yet it also leads to an increase in computational burden [30]. Hence, the determination of filter length involves a compromise between two competing factors.

To overcome this drawback, an improved adaptive MOMED (IAMOMED) is explored in this work. The improvements are listed as follows:An envelope autocorrelation function is used to automatically estimate a precise period, instead of artificially setting a period range in original MOMED;The target vector in the original MOMED is redefined to better align with real working conditions;The classical optimization algorithm known as particle swarm optimization (PSO) is employed to optimize the filter size.

The subsequent sections of this paper are organized in the following manner. Section 2 provides an overview of the basic concept of MOMED, followed by a discussion of its limitations. In Section 3, the improved MOMED is introduced. Section 4 validates the improved MOMED using simulated bearing data. Section 5 employs real datasets to further demonstrate the effectiveness of the improved MOMED method. Section 6 concludes.

## 2. Brief Review of MOMED

### 2.1. MOMED Technique

MOMED is designed to identify bearing faults featuring multiple impulses by utilizing the multi-D-norm of a filtered signal as an objective function. This multi-D-norm of a filtered signal is defined as [29]:(1)$$MDN\left(\vec{y},\vec{t}\right)=\frac{1}{\left\|\vec{t}\right\|}\frac{{\vec{t}}^{T}\vec{y}}{\left\|\vec{y}\right\|}$$
where the target vector $\vec{t}$ represents a constant vector that determines the weightings and positions of the goal impulsive components to be highlighted.

The filtered signal $\vec{y}$ is obtained by convolving the input signal $\vec{x}$ with the inverse filter coefficients $\vec{f}$.
(2)$$\vec{y}=\vec{x}*\vec{f}$$

The optimal inverse filter for MOMED is achieved by maximizing the multi–D–norm of the filtered signal:(3)$$\underset{\vec{f}}{\max}\mathrm{MDN}\left(\vec{y},\vec{t}\right)=\underset{\vec{f}}{\max}\frac{t^{T}\vec{y}}{\left\|\vec{y}\right\|}$$

The extremes of Equation (3) are obtained by taking the derivative with respect to $\vec{f}$ and computing for extremes by equating to zero. The output solutions of optimal inverse filter $\vec{f}$ are computed as follows:(4)$$\vec{f}={\left({\vec{X}}_{0}{\vec{X}}_{0}^{T}\right)}^{-1}{\vec{X}}_{0}\vec{t}$$
(5)$${\vec{X}}_{0}={\left[\begin{array}{cccccc} x_{L} & x_{L+1} & x_{L+2} & \ldots & \ldots & x_{N}\\ x_{L-1} & x_{L} & x_{L+1} & \ldots & \ldots & x_{N-1}\\ x_{L-2} & x_{L-1} & x_{L} & \ldots & \ldots & x_{N-2}\\ \vdots & \vdots & \vdots & \ddots & \ldots & \vdots \\ x_{1} & x_{2} & x_{3} & \ldots & \ldots & x_{N-L+1} \end{array}\right]}_{L\;by\;N-L+1}$$

Before applying MOMED to extract the fault–induced periodic impulses, it is necessary to consider the solution targets of the cyclic impulses, and the solution targets are obtained as:(6)$$t_{n}=P_{n}\left(T\right)=\delta_{round\left(T\right)}+\delta_{round\left(2T\right)}+\delta_{round\left(3T\right)}+\ldots +\delta_{round\left(nT\right)}$$
where $\delta_{n}$ stands for an impulse at sample *n* and *T* denotes the prior period. The final filtered signal $\vec{y}$ can be calculated using matrix operation:(7)$$\left[{\vec{y}}_{1},{\vec{y}}_{2},\cdots ,{\vec{y}}_{N}\right]={\vec{X}}_{0}^{T}{\left({\vec{X}}_{0}{\vec{X}}_{0}^{T}\right)}^{-1}{\vec{X}}_{0}\left[{\vec{t}}_{1},{\vec{t}}_{2},\cdots ,{\vec{t}}_{N}\right]$$

### 2.2. Advantages and Disadvantages of MOMED

Compared to conventional deconvolution–based methods such as minimum entropy deconvolution (MED) and maximum correlated kurtosis deconvolution (MCKD), the most significant improvement of MOMED lies in its provision of a non-iterative optimal solution for selecting a deconvolution filter. More impressively, MOMED employs a target vector $\vec{t}$ to locate each impulse and then recovers periodic impulse trains. In the presence of Gaussian background noise alone, this property may not be immediately obvious, but when random impulsive noise appears in vibration signals, the significance of this property is amplified. However, a period range as prior knowledge has to be pre–defined, which poses constraints to its practical applications. Moreover, in the original MOMED, the target vectors always initiate from an impulse. In real working environments, such situations are rare as the first impulse seldom happens to be located at the starting point of the sampling [26]. The performance of MOMED is also affected by the selection of filter length.

## 3. Improved Adaptive Multipoint Optimal Minimum Entropy Deconvolution

### 3.1. Estimation of the Period

As discussed previously, an impressive feature of MOMEDA lies in its utilization of a time target vector to achieve deconvolution to enhance the locations of the impulses, including fault information. The fault periods of a damaged bearing exhibit regularity, thus enabling the time target vectors to accurately pinpoint the unique locations of an impulse train. For example, $\vec{t}=[0\;0\;0\;0\;1\;2\;0\;0\;1\;2\;2\;0\;1]$, where $\vec{t}$ represents a target vector, and 1 indicates the location of each goal impulse, while 2 indicates the location of each random impulse. It can be observed from the target vector that one fault–related impulse occurs at sampling point *n* = 5, 9, and 13, while three random impulses occur at sampling points *n* = 6, 10, and 11. As a result, the interval (period) between two neighboring fault–induced impulses is *T_s_* = 4. The key to detecting cyclic impulses from random impulses using the MOMEDA method lies in accurately determining the period *T_s_*, which is calculated by measuring the number of sampling points between two consecutive cyclic impulses.

Theoretically, achieving the period *T_s_* is a simple task, whose expression is calculated as:(8)$$T_{s}=F_{s}\cdot T$$
where *F_s_* is the sampling rate, *T* is the period of fault-related impulses, and *T* = 1/fault characteristic frequency (FCF). Although the theoretical FCF can be obtained by calculating the geometric parameters of a rolling bearing, the actual FCF may differ from the theoretical FCF due to errors in machining and installation [26]. Moreover, the acquisition of the geometric parameters might not be feasible in advance under practical conditions. Therefore, we use the envelope autocorrelation function to calculate the period *T_s_* in this study.

An optimal signal period is determined by locating the peak position in the autocorrelation function of its envelope signal [11].

Use Hilbert transform to transform the signal *x*(*t*)
(9)$$X\left(t\right)=H\left\{x\left(t\right)\right\}=\frac{1}{\pi }\int_{-\infty }^{+\infty }\frac{x\left(\tau \right)}{t-\tau }d\tau$$
and then the corresponding enveloping signal *ENV*[*x*(*t*)] is
(10)$$ENV\left[x\left(t\right)\right]=\sqrt{X^{2}\left(t\right)+x^{2}\left(t\right)}$$

Remove the mean of the enveloping signal, and we get
(11)ENV[x(t)]=ENV[x(t)]−mean{ENV[x(t)]}

The envelope autocorrelation function can be found to be
(12)$$r_{ENV\left[x\left(t\right)\right]}\left(\tau \right)=\int ENV\left[x\left(t\right)\right]ENV\left[x\left(t+\tau \right)\right]dt$$

The period *T_s_* is obtained by finding the maximum of Equation (12).

### 3.2. Rearrangement of the Target Vector

As mentioned above, the target vector $\vec{t}$ can be regarded as a collection of impulse sequences with progressively increasing intervals. The components in the target vector $\vec{t}$ can be expressed as:(13)$$\begin{array}{l} t_{nR_{1}}={\left[1\;0\;\cdots \;1\;0\;\cdots \;0\;1\;0\;\cdots \right]}_{N-L}^{T}\\ t_{nR_{2}}={\left[1\;0\;\cdots \;0\;1\;\cdots \;0\;0\;1\;\cdots \right]}_{N-L}^{T}\\ \vdots \\ t_{nR_{R}}={\left[1\;0\;\cdots \;0\;1\;\cdots \;0\;0\;1\;\cdots \right]}_{N-L}^{T} \end{array}$$

The target vector $\vec{t}$ is therefore found to be
(14)$${\vec{t}}_{n}={\left[t_{nR_{1}},t_{nR_{2}}\cdots ,t_{nR_{R-1}},t_{nR_{R}}\right]}_{N-L\times R}$$

According to Equation (13), it is found that an impulse always occurs at the beginning of a target vector $\vec{t}$. However, such a situation is rare in practical scenarios since the first impulse seldom coincides with the sampling start time. Therefore, Equation (13) is rearranged as:(15)$$\begin{array}{l} {\vec{t}}_{nR_{1}}={\left[0\;0\;\cdots \;1\;0\;\cdots \;0\;1\;0\;\cdots \right]}_{N}^{T}\\ {\vec{t}}_{nR_{2}}={\left[0\;0\;\cdots \;0\;1\;\cdots \;0\;0\;1\;\cdots \right]}_{N}^{T}\\ \vdots \\ {\vec{t}}_{nR_{R}}={\left[0\;0\;\cdots \;0\;1\;\cdots \;0\;0\;1\;\cdots \right]}_{N}^{T} \end{array}$$
and the target vector becomes
(16)$${\vec{t}}_{n}={\left[t_{nR_{1}},t_{nR_{2}},\cdots ,t_{nR_{R-1}},t_{nR_{R}}\right]}_{N\times R}$$

### 3.3. Selection of Filter Length

Classical particle swarm optimization (PSO) has been extensively available in the field of fault diagnosis [31,32,33]. PSO has a straightforward principle and mechanism, which leads to fast convergence of the operation process. Additionally, it exhibits excellent convergence performance and global search capabilities. Further details regarding PSO can be referred to [34]. In this subsection, our attention is directed towards the establishment of the objective function. In literature, kurtosis is a widely used statistical indicator due to its sensitivity to the fault–induced cyclic impulse trains. However, kurtosis only considers the impulsiveness while ignoring cyclostationarity, so kurtosis may measure the impulsiveness of random impulses rather than that of cyclic impulses when random impulses exist in vibration signals. To this end, a new optimization objective is created by incorporating the Gini index and the envelope spectrum kurtosis (ESK) to address this issue. Recent studies have suggested that the Gini index exhibits greater robustness to outliers compared to the kurtosis [35]. The ESK index is mainly used for the measurement and evaluation of cyclostationarity [36]. Thus, the proposed indicator fully considers the characteristics in impulsive noise environments. The alternative fitness function is found to:(17)GESK=Gini/ESK
$$\left\{ \begin{array}{l} fitness=\arg\min_{filter\;length}\left\{GESK\right\}\\ filter\;length\in \left[100,1000\right] \end{array}\right.$$
in which
(18)$$Gini\left(x\right)=1-2\sum_{i=1}^{N}\frac{x_{\left(i\right)}}{{\left\|x\right\|}_{1}}\left(\frac{N-i+\frac{1}{2}}{N}\right)$$
where (i) denotes the new indices resulting from sorting a vector *x*; ${\left\|x\right\|}_{1}$ is the 1-norm of *x*.
(19)$$ESK=\frac{\sum_{l=1}^{l}{\left|SE\left(l\right)\right|}^{4}}{{\left(\sum_{l=1}^{l}{\left|SE\left(l\right)\right|}^{2}\right)}^{2}}$$
where *SE* is the envelope spectrum and *l* is the sampling number.

### 3.4. Improved MOMED Technique

According to the aforementioned discussion, an IAMOMED method is proposed for enhancing vibration signals in the presence of random impulsive noise. The envelope autocorrelation function is utilized for automatic estimation of the period *T_s_*. In order to be more in line with the practical applications, the target vector is also rearranged. The details of the improved adaptive MOMED (IAMOMED) are given below:

Step 1: Extract vibration signals from the rotating machine and specify an input parameter of window function ***w*** as [5 5 5 5 5].

Step 2: Transform the signal using the Hilbert transform and calculate the envelope autocorrelation function of the transformed signal to obtain the period *T_s_*.

Step 3: Determine the appropriate range for the filter length *L* in accordance with Equation (17).

Step 4: Utilize PSO to select an optimal the length of the filter.

Step 5: Obtain the optimal filter length *L*.

Step 6: Identify the fault-related cyclic impulses from the random impulsive noise and other interfering components using the IAMOMED with optimal filter length *L* and the period *T_s_* derived from Equation (12).

Step 7: Obtain the deconvolved signal using Equation (7).

Step 8: Perform envelope analysis on the filtered signal.

The flowchart of IMOMED is given in Figure 1.

## 4. Simulations

### Simulated Model of Bearing Vibration Signals

To examine the effectiveness of the IMOMED method, a simulated outer-race bearing fault signal generated by a bearing fault model [26] is used in this section. This simulated bearing fault signal can be represented by a model:(20)$$y\left(t\right)=x_{1}\left(t\right)+x_{2}\left(t\right)+\eta \left(t\right)+n\left(t\right)$$

In Equation (20), the first sub-signal *x*_1_(*t*) denotes the fault–related repetitive impulses, whose mathematical model is found to:(21)$$x_{1}\left(t\right)=\sum_{i=1}^{M}A_{i}e^{-D_{a}\left(t-T_{i}-\delta_{i}\right)}\sin\left(2\pi f_{r}\left(t-T_{i}-\delta_{i}\right)+\varphi \right)$$
where *A_i_* = 0.8 denotes the amplitude, *T_i_* = 0.01 s denotes the fault period, and *δ_i_* denotes the random jitter (1–2% of *T_i_*). *f_r_* = 3 kHz, *D_a_* = 420 Hz, and *ψ* = 0 represent the resonance frequency, damping coefficient, and phase angle, respectively.

The second sub–signal *x*_2_(*t*) simulates the interfering components from adjacent elements:(22)$$x_{2}\left(t\right)=\sum_{j}^{J}B_{j}\sin\left(2\pi f_{j}t+\alpha_{j}\right)$$
in which *B* is the amplitude, and *f_j_* and *α_j_* are the frequency and phase, respectively. The simulated signal is corrupted by two interfering components. The amplitudes, frequencies, and phases of the two vibration components are set to *B*_1_ = *B*_2_ = 1, *f*_1_ = 4 Hz, *f*_2_ = 10 Hz, *α*_1_ = *π*/2, and *α*_2_ = *π*/3.

The third and fourth parts, *η*(*t*) and *n*(*t*), respectively, stand for the non–Gaussian random impulsive noise and Gaussian background noise.

In this case, the simulated bearing fault signal exhibits a signal–to–noise ratio (SNR) of −5 dB and a signal–to–interference ratio (SIR) of −25 dB. The fault period *T_i_* is set to 0.01 s, that is, fault characteristic frequency (FCF) is equal to 1/*T_i_* = 100 Hz. The sampling rate is 20 kHz.

The complex signal mixture and its constituents are shown in Figure 2.

The simulated signal mixture is depicted in Figure 3. As evident in this figure, the three different kinds of interfering components exert a significant influence on the fault–induced impulses, making the periodic impulses indiscernible. Additionally, the energy of additive non−Gaussian impulsive noise is significantly higher than that of the cyclic impulses.

The IAMOMED method is used to highlight repetitive impulses. The period *T_s_* is calculated to be 200 using the envelope autocorrelation function in IAMOMED. The convergence curve of PSO in search for the minimum *GESK* is displayed in Figure 4. The resulting optimal length *L* for the filter is 565.

Figure 5 displays the waveform of the filtered signal using IAMOMED with *L* = 565 and its corresponding Hilbert envelope spectrum. As expected, the proposed IAMOMED method effectively eliminates the interfering components *x*_2_(*t*) and random impulsive noise as well as attenuating the background noise simultaneously, thereby rendering cyclic impulses distinguishable. The envelope spectrum also exhibits FCF and its associated harmonics at 2FCF, 3FCF, 4FCF, and 5FCF.

Therefore, combining PSO with MOMED to obtain the optimal filter length can not only improve fault feature extraction but also save computing resources.

## 5. Experimental Study

The datasets provided by the Case Western Reserve University (CWRU) [37] have now become a benchmark for bearing diagnosis. Thus, the data will be employed to assess the efficacy of the proposed IAMOMED method. Some related methods for coping with random impulsive noise in the literature are also used in comparison to this proposed method.

Figure 6 exhibits the basic test bench. Since the test bench has been extensively introduced in numerous studies, it will not be reiterated here. Please refer to [37] for further details. The fault signals were collected for a duration of 10 s, with sampling frequencies of 12 and 48 kHz employed in each case.

### 5.1. Case 1: Outer Ring Fault Extraction

The vibration data from the data serial number OR014@6_3 is utilized in this case. The sampling rate is 12 kHz, while the rotational speed is 1723 r/min. The characteristic frequency of the outer ring fault is calculated to be *f*_o_ = 102.9 Hz. The width of the pitting in the outer ring measures 0.014 inches. The fault signal measured from the outer ring and its envelope spectrum is displayed in Figure 7.

It is clearly seen that the raw signal contains some impulsive components with large amplitude and intensive background noise. The fault repetitive impulsive components are entirely masked by the interference. Moreover, the envelope spectrum of the unprocessed signal fails to identify the fault characteristic frequency (FCF) at *f*_o_ and its associated harmonics. Accordingly, the direct application of envelope analysis for detecting the FCF at *f*_o_ from the heavily corrupted signal is not feasible. Pre–processing of the raw data is an indispensable step in this case.

The proposed IAMOMED is used to highlight cyclic impulses. As before, the period *T_s_* is determined to be 118.4 using the envelope autocorrelation function. Subsequently, PSO is employed to select the filter length *L*. Figure 8 plots the convergence curve of PSO for minimizing *GESK*. The optimal filter length *L* = 960 is obtained.

The signal filtered using IAMOMED and the envelope spectrum of the purified signal are illustrated in Figure 9, where the cyclic impulsive components containing fault information are prominently highlighted. The FCF at *f*_o_ and its first five harmonics are also dominant in the envelope spectrum of the filtered signal. The result indicates the occurrence of an outer ring fault in the bearing and confirms the effectiveness of the proposed approach.

For comparison, two different informative frequency band (IFB) selection methods, which are a kind of commonly used method for dealing with random impulsive noise, are applied. The IFB selection tools share a similar advantage with the IAMOMED method in that they are robust against random impulsive noise, but their basic principles are fundamentally different. The IFB selection tools primarily utilize different statistical indicators to optimize the selection of accurate IFB for subsequent envelope spectrum calculations.

The Autogram method [27] is the first IFB selection method for comparison. The Autogram of the raw signal is illustrated in Figure 10.

According to the Autogram, it is found that the values of the two parameters (bandwidth *Bw* and center frequency *f_c_*) are 187.5 Hz and 5906.25 Hz, respectively. A band–pass filter is designed with the two parameters. The signal filtered by employing the band–pass filter and its envelope spectrum are depicted in Figure 11.

As is evident in Figure 11a, the band–pass filter designed using *Bw* = 187.5 Hz and *f_c_* = 5906.25 attenuates the energy of the random impulsive noise. Nevertheless, the FCF at *f*_o_ = 102.9 Hz and relevant harmonics like 2 *f*_o_, 3*f*_o_, and 4 *f*_o_ are not discernible in the envelope spectrum.

The second IFB selector is the Accugram method [38]. Figure 12 displays the Accugram of the raw signal.

As before, a band−pass filter is designed using *Bw* = 375 Hz and *f_c_* = 2062.5 Hz. The band−pass filtered signal and envelope spectrum of the filtered signal are shown in Figure 13. It is found in Figure 13a that a random impulse still remains in the filtered signal. Due to the removal of a large amount of random impulsive noise, the FCF at *f*_o_ can be recognized in the envelope spectrum. However, compared to the result in Figure 9b, this envelope spectrum only shows the FCF without its harmonics. Accordingly, the Accugram−based method is inferior to the IAMOMED−based method.

According to the two comparative tests, we find that even under the same working conditions, there are differences in the estimated parameters such as bandwidth and center frequency. This is because the IFB selection methods mainly depend on different statistical indicators to optimize the two important parameters. Once the working conditions that a statistical indicator is suitable for have been changed, this indicator may no longer be available. In contrast to the IFB selectors, the IAMOMED method does not rely on any statistical indicator but instead constructs a target vector related to fault−induced repetitive impulses. This means that the IAMOMED method is capable of adapting to different working conditions.

Finally, a classical method to attenuate background noise belonging to the family of deconvolution tools named minimum entropy deconvolution (MED) is employed in comparison with IAMOMED. Similarly to the MOMED method, the value of filter length exerts a significant influence on the performance of MED. Thus, PSO is also used to optimize MED to select an optimal filter length *L*. The convergence curve of PSO in search of the minimum *GESK* is shown in Figure 14. The resulting optimal length *L* is 55.

The filtered signal using optimized MED is displayed in Figure 15. As can be found, the MED with optimal filter length cannot remove the random impulsive noise completely. A prominent impulsive component remains in the filtered signal. The envelope spectrum also fails to exhibit the *f*_o_ at 102.9 Hz, except for some irrelevant frequencies. Therefore, the commonly used method to filter out the background noise is susceptible to random impulsive components, leading to poor fault detection performance.

### 5.2. Case 2: Inner Ring Fault Extraction

The real data used in the second case of inner ring fault detection are from [39]. The test rig is a vibrating screen, as shown in Figure 16. Vibrating screens are utilized in the mining industry for material screening, impurity removal, and material classification. When the vibrating screen is running, the material stream continuously falls onto the screen deck, leading to time-varying impacts that manifest as random impulsive noise in the vibration signals. Thus, the fault signals recorded from the rolling bearings on the vibrating screen are heavily contaminated with background noise and random impulsive noise. In addition, the raw signal will also contain a deterministic component resulting from the exciting force generated by the eccentric blocks. Accordingly, the fault vibration signals collected from the vibrating screen are more complex than those acquired from CWRU.

In this test, the sampling frequency is 20 kHz. The fault signals were recorded at a rotating speed of 960 rpm. The characteristic frequency of the inner ring fault is *f_i_* = 145.85 Hz.

Figure 17 displays the inner ring fault signal. As mentioned above, we can obviously see that three different components, including random impulsive noise, a sinusoid−like deterministic signal, and background noise, dominate the fault signal measured from the inner ring fault. The envelope spectrum cannot also recognize the FCF at *f_i_* and its harmonics.

The IAMOMED method is now used to highlight the fault−related periodic impulsive components. Similarly, the period *T_s_* is calculated to be 136.8 using the envelope autocorrelation function. The PSO is then employed to obtain an optimal filter length *L.* Figure 18 shows the PSO convergence curve. The corresponding optimal filter length is *L* = 1200.

The IAMOMED method with *T_s_* = 136.8 and *L* = 1200 is utilized to process the raw signal. The filtered signal and its envelope spectrum are shown in Figure 19. As can be seen in Figure 19a, the IAMOMED method cancels out the random impulsive noise and the deterministic component as well as highlighting the fault cyclic impulses simultaneously. The FCF at *f_i_* and its multiple harmonics can be clearly distinguished in the envelope spectrum.

As before, a nonlinear filtering method [40] is used for comparison with the proposed method. The nonlinear filtering method is developed on the basis of the alpha−stable distribution, which is well suited to describing impulsive components with high amplitude and random distribution. Therefore, the nonlinear filter (stable filter for short) is capable of removing random impulsive noise. Both the stable filter and IAMOMED are filtering algorithms, but the former primarily focuses on random impulsive components while the latter concentrates more on cyclic impulsive components. In other words, the stable filter can directly remove random impulsive noise, while IAMOMED identifies fault−induced cyclic impulses by locating each impulse to indirectly reduce the impact of random impulsive noise.

Figure 20 shows the signal purified by the stable filtering tool. Comparing Figure 20 with Figure 19a illustrates that, although the stable filter can effectively eliminate the random impulsive noise, it fails to remove the deterministic component. Thus, the filtered signal is further analyzed using the classical variational mode decomposition (VMD). The VMD method can decompose a signal into a group of mono-components. Therefore, the vibration interference can be separated from the raw signal mixture using VMD. Furthermore, VMD acts as filter bank in nature, enabling it to remove a lot of background noise [41].

Figure 21 depicts the decomposed intrinsic mode functions (IMFs). It is found from IMF1 that the deterministic component is separated from the filtered signal. However, the decomposed IMF2 and IMF3 still contain some impulsive noise with low amplitude. Thus, similarly to the MED method, VMD used to remove common background noise is also incapable of canceling random impulsive noise.

Finally, the IMF2 is applied to the envelope analysis. The envelope spectrum is displayed in Figure 22.

As is evident from this figure, the resulting envelope spectrum can detect the FCF at *f_i_*, but the FCF is not easy to identify due to the high amplitude of other frequencies surrounding the FCF. Moreover, the envelope spectrum solely illustrates the second harmonics at *2f_i_* and the fourth harmonics at 4*f_i_*.

According to the test, we can observe that the stable filter can effectively cancel out intensive random impulsive components, but it lacks the ability to remove deterministic components. Therefore, other filtering methods must be combined with the stable filter when vibration signals are interfered with by such components. Additionally, the ultimate extraction outcome may not meet expectations.

## 6. Conclusions

In this study, an improved adaptive multipoint optimal minimum entropy deconvolution (IAMOMED) is proposed for highlighting fault-related cyclic impulses in the presence of intensive random impulsive noise. The superiority and effectiveness of IAMOMEDA is examined using synthetic data and further demonstrated using real-world vibration signals from CWUR and a vibrating screen. The comparative experiments that are conducted in this work result in the conclusions below.

Considering the particularity of random impulsive noise, an objective function suitable for impulsive noise environments is established. The particle swarm optimization (PSO) can better optimize a filter length *L* using the proposed objective function in the impulsive noise environments. The IAMOMED method based on PSO is capable of precisely extracting the fault features, avoiding the blind selection of the filter length.The envelope autocorrelation function is utilized to estimate the period *T_s_* instead of setting the search range. Furthermore, the target vector in the original MOMED is rearranged to enhance its practical applicability.Compared to the original MOMED and other methods used to deal with random impulsive noise, the IAMOMED has a higher performance in highlighting cyclic impulses. It shows robustness in the face of changing working conditions.

However, combining MOMED with PSO for the filter length optimization results in a heavy computational burden. This flaw may impede the applications of the proposed method. The issue will be given greater emphasis in our future work.

## Figures and Tables

**Figure 1 entropy-25-01171-f001:**
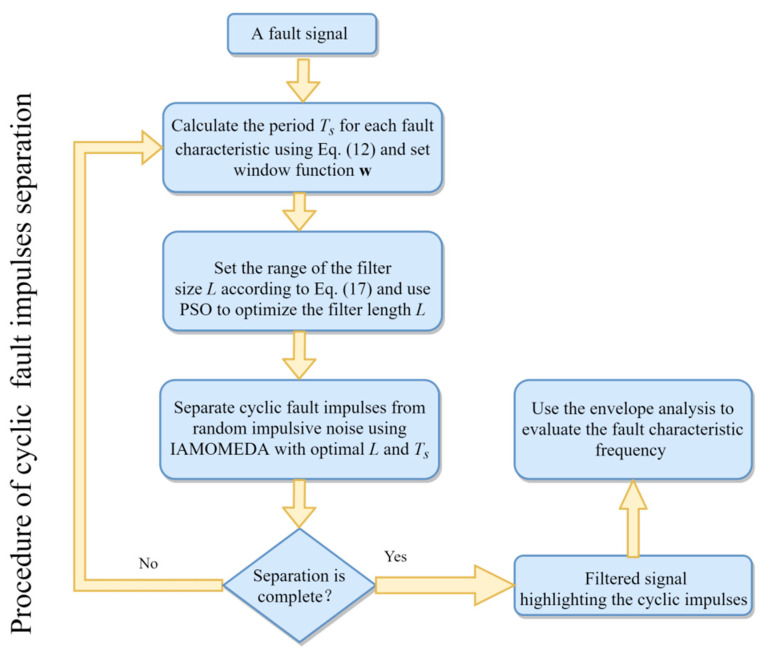
Flowchart of IAMOMED.

**Figure 2 entropy-25-01171-f002:**
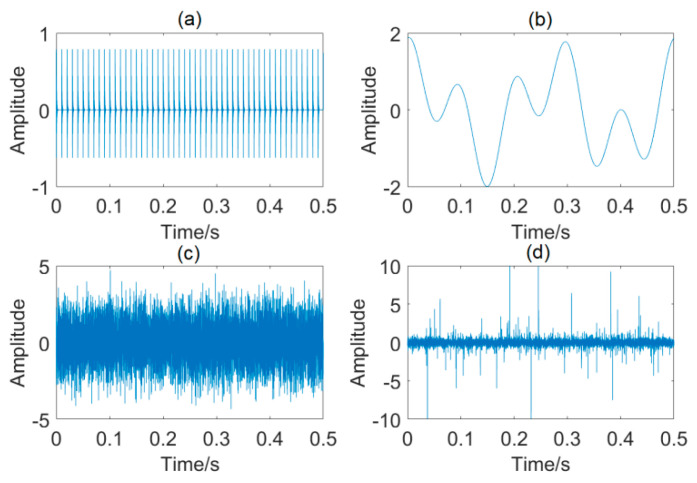
Constituents of the signal mixture: (**a**) fault−related cyclic impulses; (**b**) interfering components; (**c**) Gaussian background noise; (**d**) non−Gaussian random impulsive noise.

**Figure 3 entropy-25-01171-f003:**
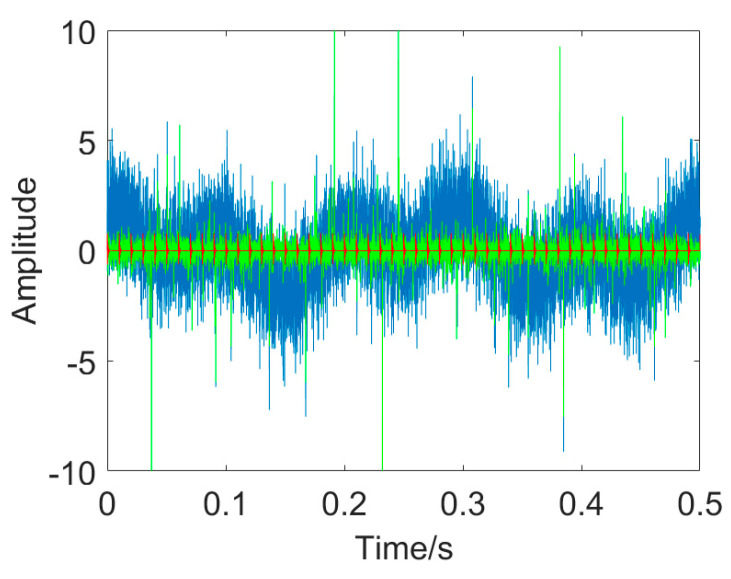
Signal mixture (The blue line denotes the background noise, the green line denotes random impulses, and the red line denotes cyclic impulses indicating faults).

**Figure 4 entropy-25-01171-f004:**
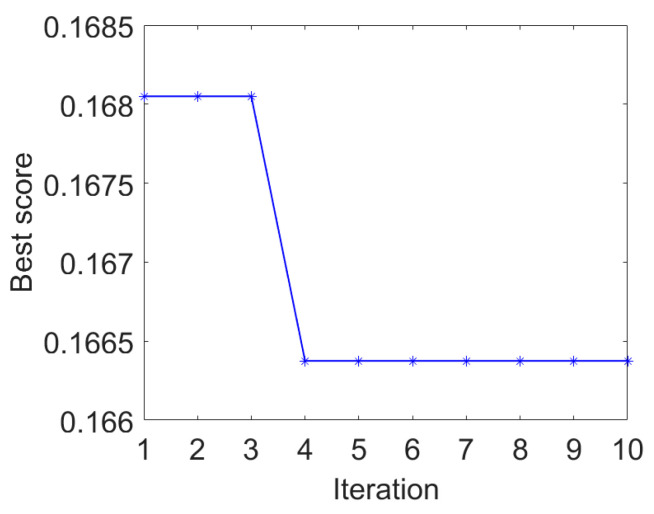
PSO convergence curve.

**Figure 5 entropy-25-01171-f005:**
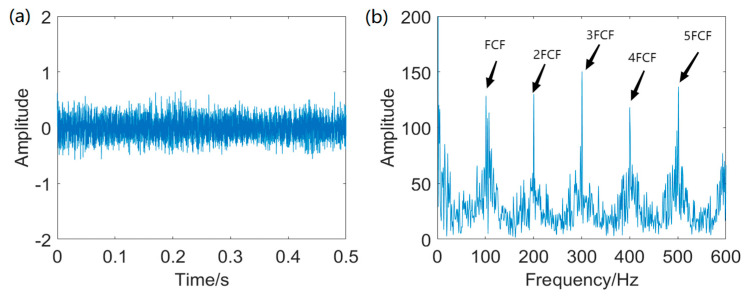
Filtered signal using IAMOMED and envelope spectrum: (**a**) filtered signal and (**b**) envelope spectrum of the filtered signal.

**Figure 6 entropy-25-01171-f006:**
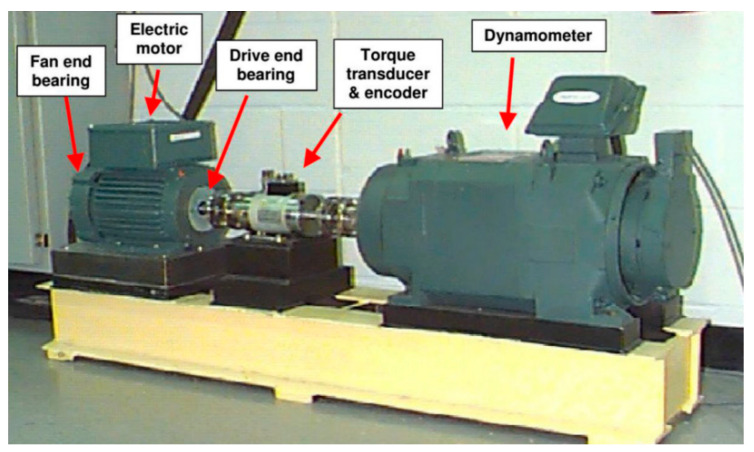
CWRU test rig.

**Figure 7 entropy-25-01171-f007:**
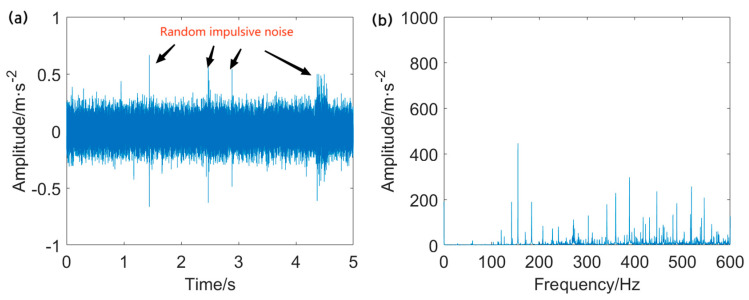
Outer ring fault signal and envelope spectrum: (**a**) vibration signal extracted from outer ring and (**b**) envelope spectrum of the outer ring fault signal.

**Figure 8 entropy-25-01171-f008:**
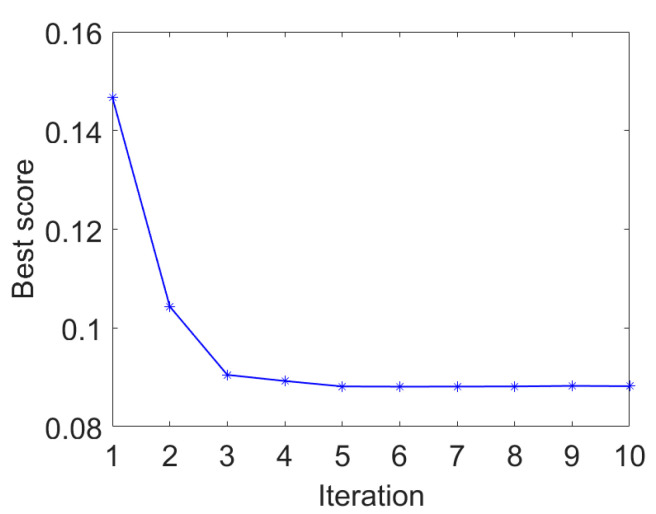
PSO convergence curve.

**Figure 9 entropy-25-01171-f009:**
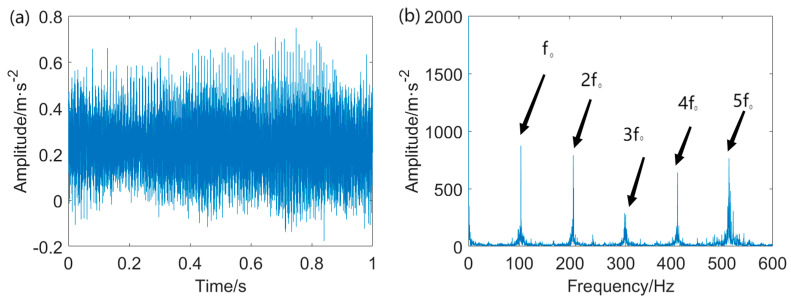
The result provided by IAMOMED: (**a**) waveform of the purified signal using IAMOMED and (**b**) envelope spectrum of the IAMOMED purified signal.

**Figure 10 entropy-25-01171-f010:**
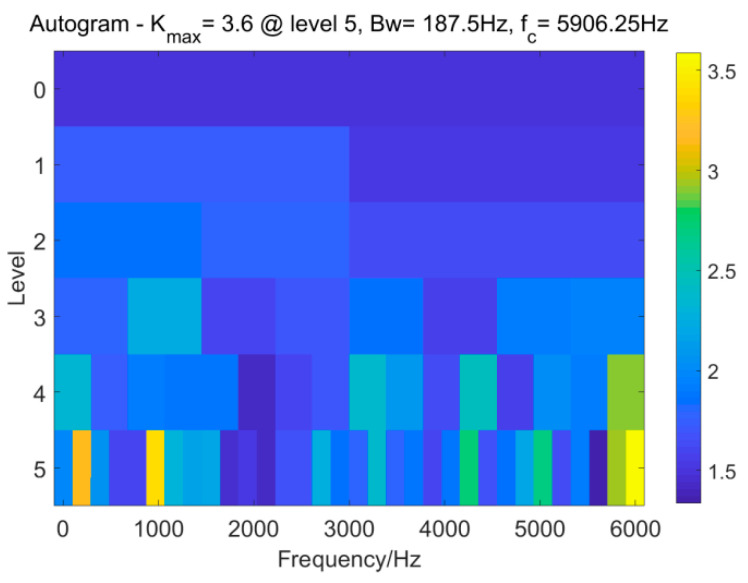
Autogram.

**Figure 11 entropy-25-01171-f011:**
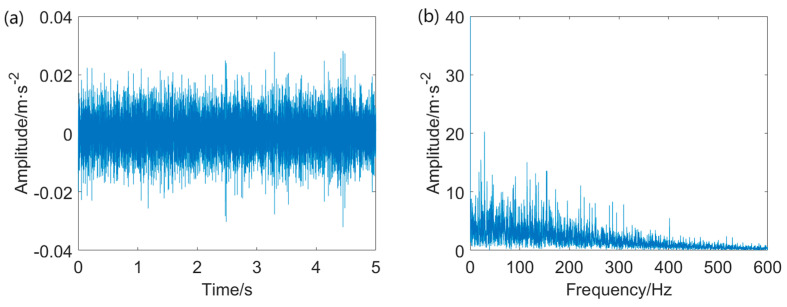
The result provided by the Autogram−based method: (**a**) band−pass filtered signal and (**b**) envelope spectrum of the band−pass filtered signal.

**Figure 12 entropy-25-01171-f012:**
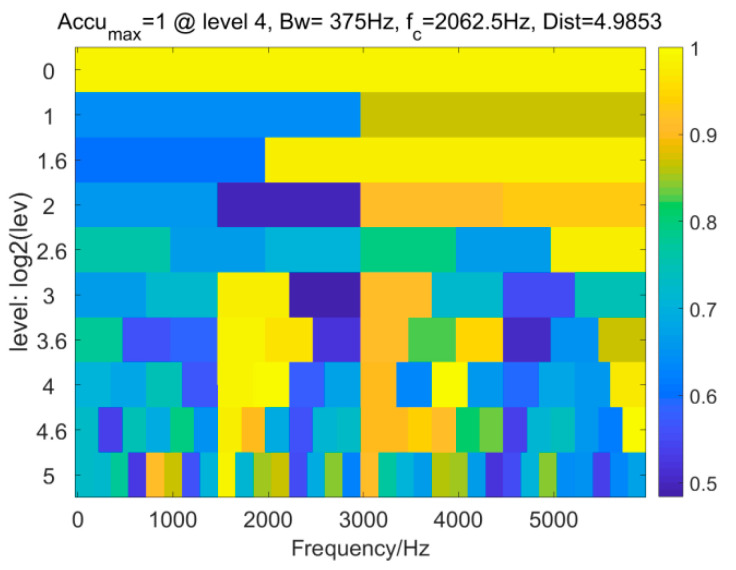
Accugram.

**Figure 13 entropy-25-01171-f013:**
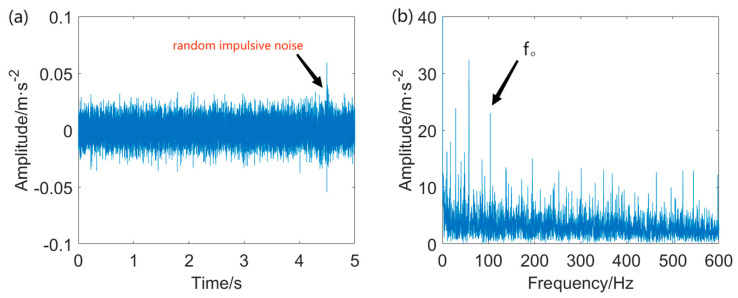
The result obtained by the Accugram−based method: (**a**) waveform of the filtered signal and (**b**) envelope spectrum of the filtered signal.

**Figure 14 entropy-25-01171-f014:**
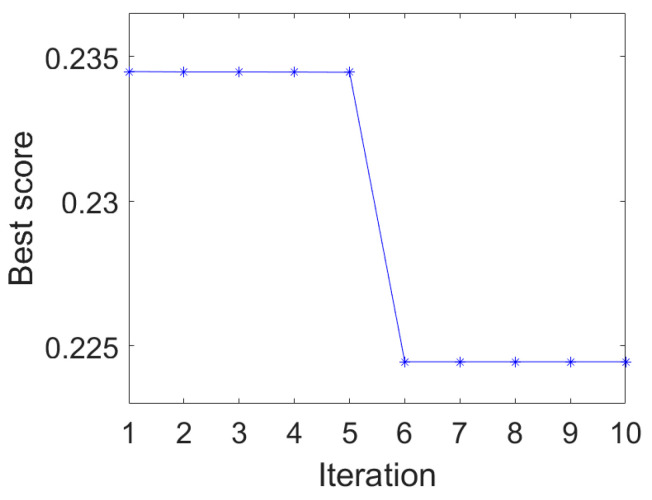
PSO convergence curve for optimizing MED.

**Figure 15 entropy-25-01171-f015:**
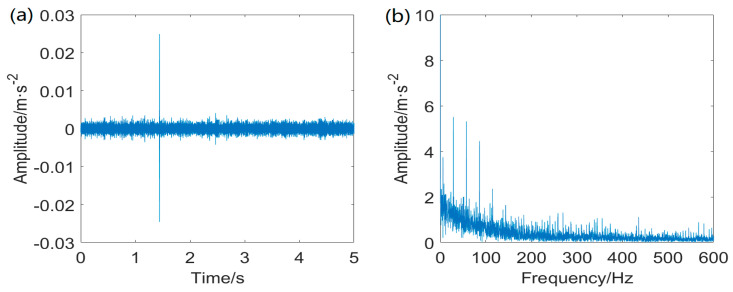
The result obtained using MED−based method: (**a**) waveform of the MED filtered signal and (**b**) envelope spectrum of the MED filtered signal.

**Figure 16 entropy-25-01171-f016:**
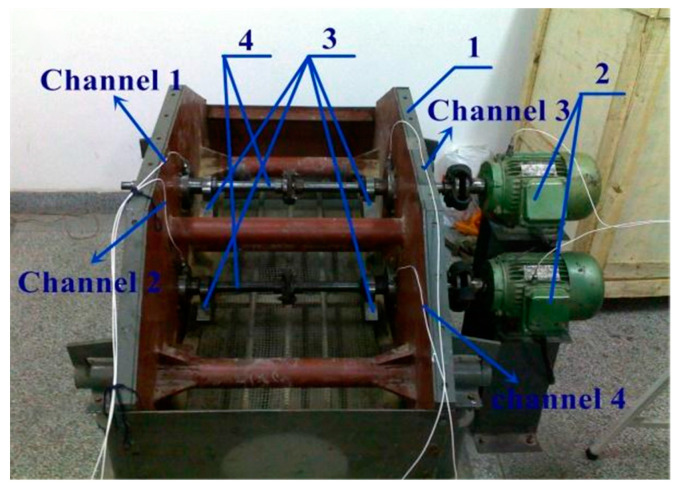
Vibrating screen test bench: (1) screen box; (2) motor; (3) eccentric blocks; and (4) transmission shafts.

**Figure 17 entropy-25-01171-f017:**
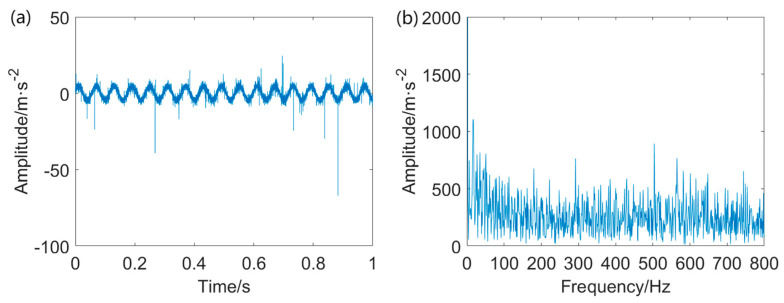
Raw inner ring fault signal: (**a**) waveform of the inner ring fault signal and (**b**) envelope spectrum of the inner ring fault signal.

**Figure 18 entropy-25-01171-f018:**
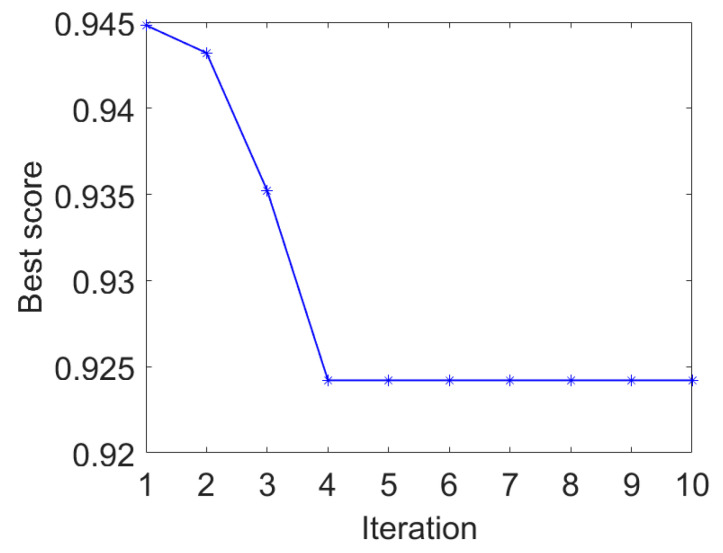
PSO convergence curve.

**Figure 19 entropy-25-01171-f019:**
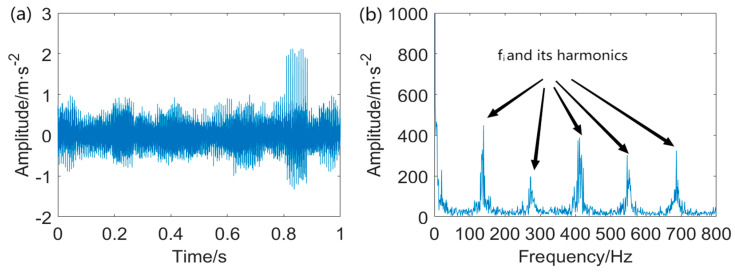
The result provided by the IAMOMED method: (**a**) waveform of the IAMOMED filtered signal and (**b**) envelope spectrum of the IAMOMED filtered signal.

**Figure 20 entropy-25-01171-f020:**
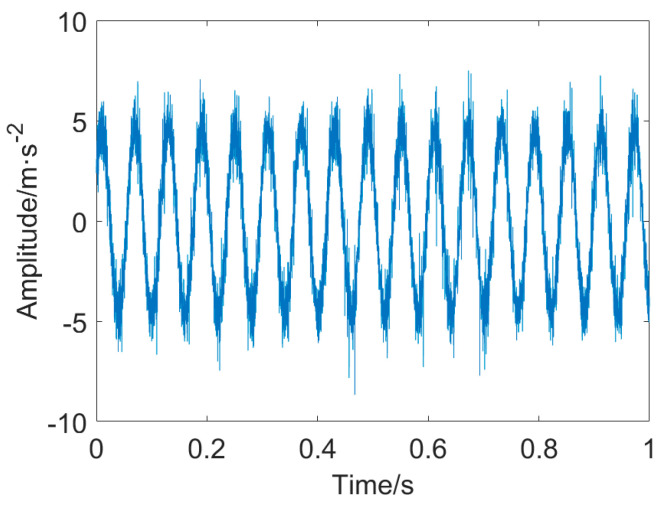
Filtered signal using the stable filter.

**Figure 21 entropy-25-01171-f021:**
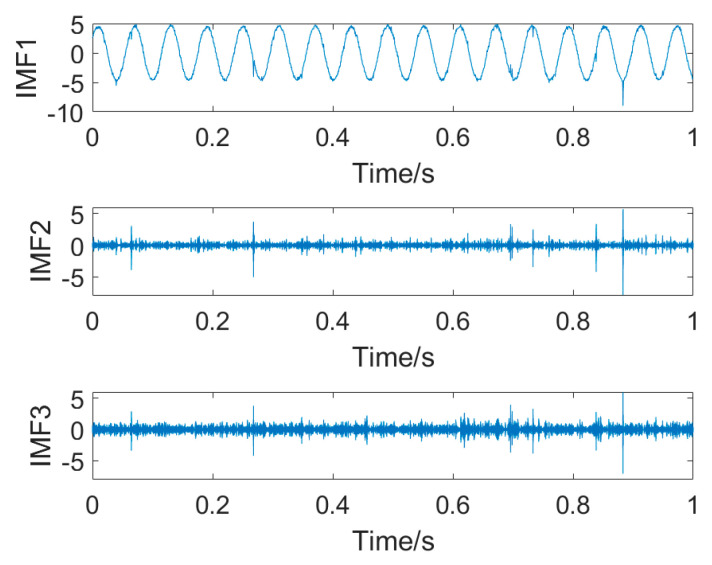
Decomposed IMFs.

**Figure 22 entropy-25-01171-f022:**
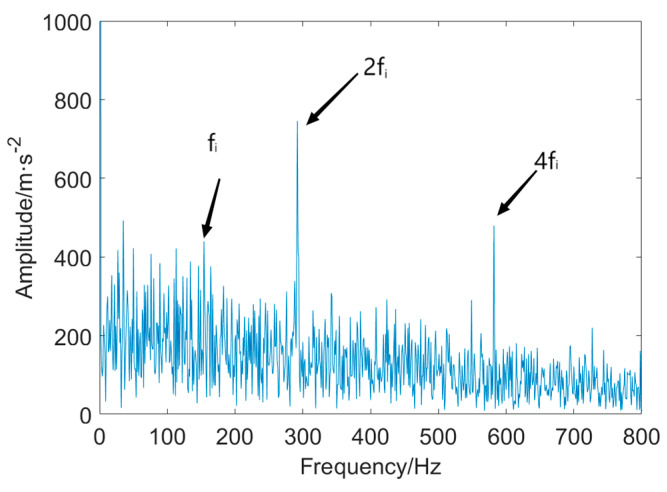
Decomposed IMFs.

## Data Availability

Not applicable.

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
