# Peer review of "Improved Adaptive Multipoint Optimal Minimum Entropy Deconvolution and Application on Bearing Fault Detection in Random Impulsive Noise Environments"

_entropy, 2023, doi:10.3390/e25081171_

Round 1
Reviewer 1 Report
The authors have proposed an improved adaptive multipoint optimal minimum entropy deconvolution approach to estimate the bearing fault-induced period. The paper was well written and thoroughly investigated. For clarity of the manuscript, this reviewer has the following comments:
(1)3.1 Estimation of the period - It can be observed that...
The occurrences of the fault-related and random impulses are unable to match with sampling points. Please recheck it.
(2)4.1 Simulated model of bearing vibration signals
Please check the two phases of the second sub-signal x2(t).
What are the signal-to-noise ratio (SNR) and the signal-to-interference ratio (SIR) of the simulated bearing fault signal?
(3) Why is not the optimal filter length (see Fig.4) being used in Fig.5?
(4) Conclusions: The future scope of the work should be provided.
Author Response
Dear Editors and Reviewer#1:
Thank you for your comments concerning our manuscript. Those comments are all valuable and very helpful for revising and improving our paper, as well as the important guiding significance to our research. We have studied the comments carefully and have made corrections which we hope meet with approval. Revised portions are marked in red in the paper. The main corrections in the paper and the responses to the reviewer’s comments are as follows:
(1)3.1 Estimation of the period - It can be observed that...
The occurrences of the fault-related and random impulses are unable to match with sampling points. Please recheck it.
Response: Thank you so much for bringing this mistake to my attention. This mistake has been corrected. The corrected content is as follows:
It can be observed from the target vector that one fault-related impulse occurs at sampling point n = 5, 9, and 13, while three random impulses occur at sampling points n = 6, 10, and 11 respectively.
(2)4.1 Simulated model of bearing vibration signals
Please check the two phases of the second sub-signal x2(t).
What are the signal-to-noise ratio (SNR) and the signal-to-interference ratio (SIR) of the simulated bearing fault signal?
Response: Thank you so much for pointing out this carelessness. This mistake has been corrected. Besides, the SNR and SIR of the simulated bearing fault signal have been added. The corrected content is as follows:
The amplitudes, frequencies, and phases of the two vibration components are set to B1 = B2 = 1, f1 = 4 Hz, f2 = 10 Hz, α1 = π/2, α2 = π/3.
In this case, the simulated bearing fault signal exhibits a signal-to-noise ratio (SNR) of -5 dB and a signal-to-interference ratio (SIR) of -25 dB.
- Why is not the optimal filter length(see Fig.4) being used in Fig.5?
Response: We apologize for the carelessness again. The results in Fig.5 is obtained using the optimal filter length L=565. Due to the carelessness, the optimal filter length was erroneously designated as L=665.
- Conclusions: The future scope of the work should be provided.
Response: Thank you so much for your suggestions. The future scope of the work has been included in section of Conclusions. The related content is give below:
However, combining MOMED with PSO for the filter length optimization results in a heavy computational burden. This flaw may impede the applications of the proposed method. The issue will be given greater emphasis in our future work.

Reviewer 2 Report
1. In the abstract section, the specifics of what makes random noise different from background noise should be directly expressed at the very beginning, leading to the advantages of the algorithms studied in this article.
2. At the end of the abstract, it should be written what kind of results the algorithm of this article has compared to other algorithms.
3. In the third paragraph of the introduction, please carefully state exactly which noise is a random distribution with varying amplitude and which is a periodic distribution with uniform amplitude. So as not to cause misunderstanding to the reader.
4. Since you have improved the MOMED method, please indicate in the introduction the current state of research on this algorithm in the area of random noise removal.
5. In the introduction section, it is recommended to give a separate description of the innovative points of the article. It is best to list the points.
6. Is the MOMEDA check in part 2.1 wrong?
7. It is recommended that the bolded font within the formula be changed to English format. In addition, the last item should be described after the ellipsis in formula (6). You can refer to the formulas in other articles. The size of the formulas is not the same, so please check the formulas in the whole article and revise them.
9. In 3.1, the example t=[...] can be placed directly in the body of the article.
10. In 4.1, the parameters of α1 are given twice and α2 is not written.
11. Figures 2 (a), (b), (c), (d) are labeled at the bottom of the figure. Figures 5, 6, 8, 10, 12, 14, 16, 18 are the same. Please check the labeling of figures throughout the article and revise it. Many of your figure annotations are not standardized, please correct them according to the journal format.
12. Among the simulated signals in 4.1, you use only a few MOMED methods for determining the filter length L. I don't think it makes any sense to compare this with PSO-IAMOMED. You should use PSO-optimized MOMED to compare with the PSO-optimized IAMOMED method presented in this article.
13. From the description of your experiment, I found that you used different comparison methods in the simulation signal, dataset 1 and dataset 2. Obviously, the advantages of your proposed algorithm cannot be clearly highlighted. The comparison algorithms you used are too few and the comparison algorithms should cover all the datasets. Your comparison algorithm should contain: PSO-MOMED, some methods to remove random noise and some methods to remove background noise.
Author Response
Dear Editors and Reviewer #2:
Thank you for your comments concerning our manuscript. Those comments are all valuable and very helpful for revising and improving our paper, as well as the important guiding significance to our research. We have studied the comments carefully and have made corrections which we hope meet with approval. Revised portions are marked in red in the paper. The main corrections in the paper and the responses to the reviewer’s comments are as follows:
- In the abstract section, the specifics of what makes random noise different from background noise should be directly expressed at the very beginning, leading to the advantages of the algorithms studied in this article.
Response: Thank you for your suggestions. The content showing distinctions between random impulsive noise and background noise is as follows:
Abstract: Random impulsive noise is a special kind of noise, which strong impact features and has random disturbances with large amplitude, short duration, and long intervals. This type of noise often displays non-Gaussianity while common background noise obeys Gaussian distribution. Hence, random impulsive noise greatly differs from common background noise, which renders many commonly used approaches in bearing fault diagnosis inapplicable. In this work, we explore the challenge of bearing fault detection in the presence of random impulsive noise. To deal with this issue, an improved adaptive multipoint optimal minimum entropy deconvolution (IAMOMED) is introduced. In this IAMOMED, an envelope autocorrelation function is used to automatically estimate the cyclic impulse period instead of setting an approximate period range. Besides, the target vector in the original MOMED is rearranged to enhance the practical applicability. Finally, particle swarm optimization is employed to determine the optimal filter length for selection purposes. According to these improvements, IAMOMED is more suitable for detecting bearing fault features in the case of random impulsive noise compared to the original MOMED. The simulated and real-world signals demonstrate the effectiveness and superiority.
- At the end of the abstract, it should be written what kind of results the algorithm of this article has compared to other algorithms.
Response: Thank you very much for your suggestions. The related content is given as follows:
Abstract: Random impulsive noise is a special kind of noise, which strong impact features and has random disturbances with large amplitude, short duration, and long intervals. This type of noise often displays non-Gaussianity while common background noise obeys Gaussian distribution. Hence, random impulsive noise greatly differs from common background noise, which renders many commonly used approaches in bearing fault diagnosis inapplicable. In this work, we explore the challenge of bearing fault detection in the presence of random impulsive noise. To deal with this issue, an improved adaptive multipoint optimal minimum entropy deconvolution (IAMOMED) is introduced. In this IAMOMED, an envelope autocorrelation function is used to automatically estimate the cyclic impulse period instead of setting an approximate period range. Besides, the target vector in the original MOMED is rearranged to enhance the practical applicability. Finally, particle swarm optimization is employed to determine the optimal filter length for selection purposes. According to these improvements, IAMOMED is more suitable for detecting bearing fault features in the case of random impulsive noise compared to the original MOMED. The contrast experiments demonstrate the proposed IAMOMED technique is capable of effectively identifying fault characteristics from the vibration signal with strong random impulsive noise and it can also accurately diagnose the fault types. Thus, the proposed method provides an alternative fault detection tool for rotating machinery in the presence of random impulsive noise.
- In the third paragraph of the introduction, please carefully state exactly which noise is a random distribution with varying amplitude and which is a periodic distribution with uniform amplitude. So as not to cause misunderstanding to the reader.
Response: Thank you very much for pointing out the problem. The corrected content is shown as follows:
However, a special kind of noise, namely random impulsive noise, can also appear in certain engineering problems such as telephone wires, image processing, radar systems, and underwater acoustics. Random impulsive noise is clearly different from traditional background noise. Random impulsive noise often exhibits non-Gaussianity while traditional background noise obeys Gaussian distribution. Furthermore, random impulsive noise shows strong impact features and has sudden disturbances with varying amplitude, short duration, and long intervals [17]. According to the above discussion, it is found that random impulsive noise is somewhat similar to fault-induced impulsive components. Both are comprised of a series of impulsive components except that random impulsive noise is a random distribution with varying amplitudes and fault-related impulsive components are a periodic distribution with a uniform amplitude. Unlike other engineering applications, distinguishing fault-induced cyclic impulses from random impulses poses a significant challenge in bearing fault diagnosis. Compared to Gaussian background noise, random impulsive noise can result in more severe signal corruption. Accordingly, the above-mentioned standard signal processing tools used for analyzing vibration signals may perform poorly or become ineffective [18, 19].
- Since you have improved the MOMED method, please indicate in the introduction the current state of research on this algorithm in the area of random noise removal.
Response: Thank you so much for your suggestions. The original MOMED tool has been widely available in bearing fault diagnosis. However, this algorithm is mainly used to eliminate the common background noise and is rarely used for random impulsive removal in literature. According to deep investigation, we have found that this method is a potential tool to distinguish the fault-related periodic impulses from random impulsive components. One of the innovations of this paper also includes this aspect. Accordingly, the related research on this algorithm in the area of random noise removal was not indicated. Thank you for your understanding.
5.In the introduction section, it is recommended to give a separate description of the innovative points of the article. It is best to list the points.
Response: Thank you so much for your recommendation. The separate description of the innovative points of the article is listed below:
To overcome this drawback, an improved adaptive MOMED (IAMOMED) is explored in this work. The improvements is listed as follows:
- An envelope autocorrelation function is used to automatically estimate a precise period, instead of artificially setting a period rangein original MOMED;
- The target vector in the original MOMED is redefined to better align with real working conditions;
- Theclassical optimization algorithm known as particle swarm optimization (PSO) is employed to optimize the filter size.
- Is the MOMEDA check in part 2.1 wrong?
Response: We apologize for the carelessness. The mistake has been corrected.
- It is recommended that the bolded font within the formula be changed to English format. In addition, the last item should be described after the ellipsis in formula (6). You can refer to the formulas in other articles. The size of the formulas is not the same, so please check the formulas in the whole article and revise them.
Response: Thank you very much for pointing out the issues. The formulas have been refined according to the comments.
- In 3.1, the example t=[...] can be
Response: Thank you very much for pointing out the issues. The example has been placed directly in the body of the article.
- In 4.1, the parameters of α1 are given twice and α2 is not written.
Response: Thank you so much for pointing out this carelessness. This mistake has been corrected. The corrected content is as follows:
The amplitudes, frequencies, and phases of the two vibration components are set to B1 = B2 = 1, f1 = 4 Hz, f2 = 10 Hz, α1 = π/2, α2 = π/3.
- Figures 2 (a), (b), (c), (d) are labeled at the bottom of the figure. Figures 5, 6, 8, 10, 12, 14, 16, 18 are the same. Please check the labeling of figures throughout the article and revise it. Many of your figure annotations are not standardized, please correct them according to the journal format.For
Response: We apologize for the carelessness. The mistakes have been corrected.
- Among the simulated signals in 4.1, you use only a few MOMED methods for determining the filter length L. I don't think it makes any sense to compare this with PSO-IAMOMED. You should use PSO-optimized MOMED to compare with the PSO-optimized IAMOMED method presented in this article.
- From the description of your experiment, I found that you used different comparison methods in the simulation signal, dataset 1 and dataset 2. Obviously, the advantages of your proposed algorithm cannot be clearly highlighted. The comparison algorithms you used are too few and the comparison algorithms should cover all the datasets. Your comparison algorithm should contain: PSO-MOMED, some methods to remove random noise and some methods to remove background noise.
Response: Thank you very much for your valuable suggestions. In light of comments 11 and 12, we would like to provide a consolidated response. Actually, the proposed IAMOMED method was also taken into consideration for comparison with PSO+MOMED, as suggested by the reviewer. In this paper, three improvements are implemented to the MOMED. The first two improvements, namely the utilization of an envelope autocorrelation function for automatic estimation of a precise period and redefinition of the target vector in the original MOMED, primarily offer a more convenient and accurate approach to determining period and locations of fault impulses yet not to significantly enhance the performance of the original MOMED. However, the value of the filter length exerts a great influence on the performance of the original MOMED (see Fig.6). As a result, the PSO method was used to optimize the original MOMED to determine the optimal the filter length. In other words, the difference between IAMOMED and PSO-MOMED is likely to be slight in terms of fault extraction performance if the the value of the filter length is appropriate. The purpose of comparing IAMOMED to the original MOMED with different filter lengths is to highlight the importance of PSO.
Since the real experiment 1 lacks a method to eliminate the background noise for comparison with IAMOMED, a commonly used method to remove background noise named minimum entropy deconvolution (MED) has been added to compare with IAMOMED in the real experiment 1. The filtering methods compared with IAMOMED in the experiment 2 have actually included a technique for eliminating random impulsive noise (stable filter) and another approach (VMD) for removing background noise, as suggested by reviewer. Although the VMD is mainly used to remove the vibration interference in the experiment 2, it also acts as filter bank in nature. The property of VMD has been added to the paper in order to enhance readers’ understanding.

Round 2
Reviewer 2 Report
The following two issues you did not change in the article.
1.In 4.1, the parameters of α1 are given twice and α2 is not written.
2.Figures 2 (a), (b), (c), (d) are labeled at the bottom of the figure. Figures 5, 6, 8, 10, 12, 14, 16, 18 are the same.
In addition, the article has a lot of refutation points, and there have been relatively few changes according to my requirements, such as the comparison method not moving at all. The explanation was also very farfetched.
Author Response
Dear Editors and Reviewer #2:
Thank you for your comments concerning our manuscript. Those comments are all valuable and very helpful for revising and improving our paper, as well as the important guiding significance to our research. We have studied the comments carefully and have made corrections which we hope meet with approval. Revised portions are marked in blue in the paper. The main corrections in the paper and the responses to the reviewer’s comments are as follows:
1.In 4.1, the parameters of α1 are given twice and α2 is not written.
Response: We apologize for this omission. This mistake has been corrected. The corrected content is as follows:
The amplitudes, frequencies, and phases of the two vibration components are set to B1 = B2 = 1, f1 = 4 Hz, f2 = 10 Hz, α1 = π/2, α2 = π/3.
- Figures 2 (a), (b), (c), (d) are labeled at the bottom of the figure. Figures 5, 6, 8, 10, 12, 14, 16, 18 are the same.
Response: We apologize for this omission. This mistake has been corrected.
In addition, the article has a lot of refutation points, and there have been relatively few changes according to my requirements, such as the comparison method not moving at all. The explanation was also very farfetched.
Response: We apologize that the previous response failed to meet the reviewer's requirements. A detailed explanation about the comparison method is provided here.
In simulations section, we have excluded the section of comparing IAMOMED with the original MOMED with different filter length according to the reviewer’s previous comments.
The IAMOMED method was compared with three different methods, namely Autogram-based, Accugram-based, and MED-based methods, in Case 1 of outer ring fault extraction. The Autogram-based and Accugram-based methods were specifically designed to cope with random impulsive noise like the proposed IAMOMED while the MED-based method is a classical tool to remove background noise in order to highlight the cyclic impulses.
In Case 2 of inner ring fault extraction, a hybrid approach combining stable filter (an impulsive noise removal tool) with VMD (a technique for eliminating background noise and vibration interference was employed to compare to IAMOMED.
According to the results of the experiments, neither the individual methods (Autogram-based, Accugram-based, and MED-based methods) nor the combined method (stable filter+VMD) is as effective as the proposed IAMOMED.
